# Laser Powder Bed Fusion of K418 Superalloy: Process, Microstructure, Texture Feature, and Mechanical Property

**Zhen Chen** [1,*], **Pei Wei** [1], **Hanfeng Chen** [2], **Xinggang Chen** [2], **Yi Ruan** [3], **Wenzheng Zhou** [2] **and Sujun Lu** [2,*]

1   State Key Laboratory of Manufacturing System Engineering, School of Mechanical Engineering, Xi'an Jiaotong University, Xi'an 710049, China; weipei@xjtu.edu.cn
2   State Key Laboratory of Nickel and Cobalt Resources Comprehensive Utilization, Jinchang 624199, China; chenhanfeng@jnmc.com (H.C.); chenxinggang@jnmc.com (X.C.); zhouwenzheng@jnmc.com (W.Z.)
3   Jinchuan Group Co., Ltd., Jinchang 624199, China; ruanyi@jnmc.com
*   Correspondence: chenzhen2025@xjtu.edu.cn (Z.C.); chenguoju@jnmc.com (S.L.)

**Abstract:** Laser Powder Bed Fusion (LPBF) is one of the most promising additive manufacturing (AM) technologies using metal powders. It has been increasingly applied in variety of industrial and engineering fields, including but not limited to aviation, aerospace, nuclear energy, automobiles, medical, molding, shipping, and so on. In this work, the influence of laser process parameters on the microstructure, textural features, and their resulting effect on the macroscopic mechanical properties of LPBF-manufactured K418 samples was investigated experimentally. OM, SEM, and X-ray diffraction were used to characterize the microstructure evolution, and EBSD was used to identify the crystal texture of the as-built K418 samples. The effect relationship between process, microstructure, and properties was investigated using mechanical property testing. Furthermore, the volumetric energy density *VED* was considered as a comprehensive evaluation index to reflect the effects of the main laser process parameters on the microstructure and mechanical behavior of LPBF-manufactured K418 samples, including scanning speed *v*, laser power *P*, layer thickness *t*, and hatch space *H*. The results show that as the volumetric energy density VED increases, the microstructure morphology of the LPBF-manufactured K418 sample evolves: clustered columnar grains → coarsened columnar grains → ultrafine columnar grains, and the mechanical properties of the LPBF-manufactured K418 sample improve, owing to the ultrafine elongated columnar grains and a strong {001} <100> cubic texture.

**Keywords:** laser powder bed fusion; K418 superalloy; process parameters; microstructure; texture; mechanical property

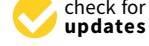



## 1. Introduction

K418 (equivalent to Inconel 713C) is a γ'-phase precipitation-strengthened nickel-based cast superalloy self-developed by China in the 1970s. Outstanding comprehensive mechanical properties and oxidation resistance at elevated temperatures, as well as good comprehensive performance, and the stability and reliability of the structure over a wide temperature range make K418 widely used in high temperatures and severe service environments, such as the hot end of the aero-engine, automobile, and shipbuilding industry [1]. However, it is not easy to produce such superalloy by traditional methods at room temperature owing to its large tool wear, large deformation at high temperature, easy hardening, and low material treatment rate. Meanwhile, K418 components have been increasingly adopting complex structures with thin-walled complexes, overhangs, or surface labyrinth channels, which would be difficult to machine by a single traditional method [2]. Therefore, without the parallel improvement of processing technology, the progress of superalloy properties is impossible [3].

In recent years, laser powder bed fusion (LPBF) has been considered a metal additively manufacturing technology most closely related to commercial mass production for the

fabrication of complex metal parts [4]. It can greatly eliminate the process constraints of traditional manufacturing, such as the complexity of parts, structural features (closed cavity, thin wall, overhang, minimum/maximum size), manufacturing processes, etc., and realize the integrated manufacturing of almost any metal parts with complex structures [5]. During LPBF processing, the material undergoes an ultrafast nonequilibrium solidification process, and the cooling rate can reach 105–109 K/s. As a result, LPBF can manufacture compact parts with excellent physical properties. Consequently, under optimal processing conditions, a nearly completely dense component with the expected ideal mechanical properties can be manufactured by LPBF [6,7]. In addition, LPBF has the potential to reduce lead time and material waste. Furthermore, it has been reported that most LPBF-manufactured components exhibit fine-grained microstructures with superior mechanical properties compared to their traditional casting and forging counterparts [8–11]. On the other hand, during LPBF processing, the ultra-high gradient temperature leads to preferential epitaxial growth of the grains, resulting in specific microstructures, textures, and mechanical properties [12]. Nickel has a face-centered cubic (fcc) crystal structure, and its alloys often exhibit columnar crystal with a cubic texture in <100>. The microstructure and crystal structure of the LPBF-manufactured samples will inevitably be affected by the complex physical metallurgical behavior and energy transmission in the molten pool, which determines the final service mechanical properties of the target components [13,14].

At present, considerable work regarding the microstructures and properties of nickel-based superalloys has been carried out, such as Inconel 625 [15], Inconel 718 [16,17], Hastelloy-X [18], Waspaloy [19], and IN738LC [20]. K418 superalloy is a critical material in the production of hot-end components for aerospace engines. It is also an indispensable material for the manufacture of high-temperature hot end components for ships, automobiles, and gas engines. It is also widely used in a variety of industries, including transportation, the petrochemical industry, energy power, medical devices, and environmental protection. It can be used to manufacture aero-engine and gas turbine turbines, working blades, guide vanes, combustion chambers, and other hot end high-temperature components operating at 600–950 °C due to its good high-temperature mechanical properties, economic outcomes and long-term structural stability, good casting performance, excellent thermal strength, thermal stability, and good resistance to mechanical fatigue and thermal fatigue. However, few studies have focused on K418 components fabricated by LPBF, which is largely due to the fact that K418 alloy, as a high-concentration $\gamma'$-strengthened superalloy, has a high content of Al and Ti elements, making the materially poor in weldability. The process window for LPBF manufacturing of K418 alloy is quite narrow and unreasonable process parameters can easily cause thermal cracks and rapidly deteriorate the performance of the material, making the engineering application of additively manufactured parts potentially risky and reliability difficult to guarantee. In addition, the cracking sensitivity and strength-plastic mismatch of LPBF manufactured K418 superalloy are the main technical bottlenecks preventing its use in high-strength and tough hot-end components. Meanwhile, due to the complexity of service conditions, the functional components manufactured by the original LPBF must be supported by numerous research data in order to play an irreplaceable role in short-term engineering practice. Therefore, it is critical to investigate the influence of process parameters on the microstructure of the parts fabricated by LPBF to improve their mechanical properties. This work aimed mainly to reveal the interactive relationship between the microstructures and mechanical properties of the K418 parts manufactured by LPBF under various processing conditions using SEM, EBSD, EDS, and X-ray diffraction.

## 2. Materials and Methods

### 2.1. Materials

Element composition of the gas-atomized K418 superalloy spherical powder is shown in Table 1, which was identified by inductively coupled plasma atomic emission spectrometry, provided by AMC Powder Metallurgy Technology Co, Ltd., Beijing, China. (NCS Testing Technology Co., Ltd., Plasma 1000, Beijing, China). Although a few small irregu-

lar satellite powders adhere to the spherical powders, the overall sphericity of the K418 powders is relatively good, as shown in Figure 1a. The particle size distribution of the K418 powder was measured by a laser diffraction particle size analyzer (Sympatec GmbH, HELOS H3185, Clausthal-Zellerfeld, Germany) as shown in Figure 1b, with the powder particle size range being 15–53 μm following a Gaussian distribution, and the average particle size is 28.59 μm. Before the experiment, the powder was dried in a vacuum oven at 100 °C for 4 h to remove the moisture of the powder and improve its flowability.

**Table 1.** Element composition of K418 Superalloy.

| Element | Ni | Cr | Al | Mo | Nb | Ti | Fe | B | C |
|---|---|---|---|---|---|---|---|---|---|
| Std. (wt%) | Balance | 11–13.5 | 5.5–6.4 | 3.8–4.8 | 1.8–2.8 | 0.5–1 | ≤1.0 | 0.008–0.02 | 0.08–0.16 |
| Actual (wt%) | Balance | 12.5 | 6.2 | 4.3 | 2.1 | 0.7 | 1.0 | 0.014 | 0.12 |

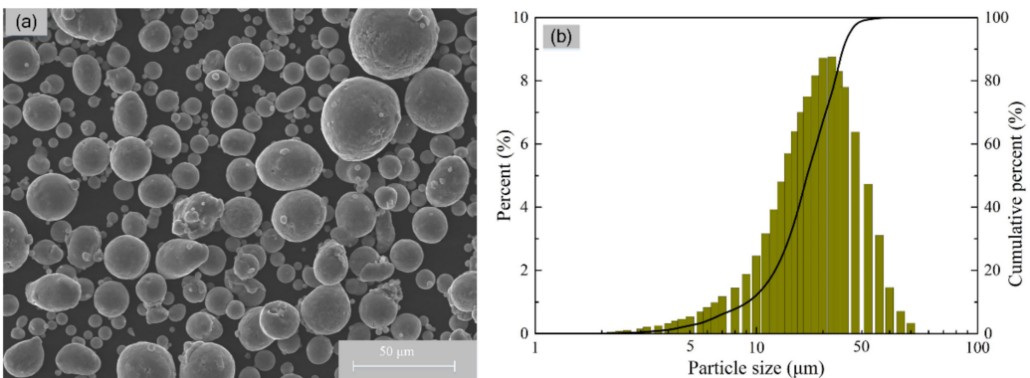

**Figure 1.** K418 powder: (**a**) SEM micrograph, (**b**) particle size distribution.

### 2.2. Specimens Preparation

In this work, a self-developed LPBF machine (RC100–N2, Xi'an Jiaotong University, China) was used for sample preparation. The equipment uses a single-mode continuous ytterbium fiber laser with a wavelength of 1064 nm and an $f$-$\theta$ lens with a spot diameter of approximately 70 μm. During the LPBF process, the oxygen content in forming chamber was controlled within 500 ppm. As shown in Table 2, cubical specimens with a size of $8 \times 8 \times 8$ mm$^3$ and dog-bone tensile specimens with a gauge length of 25 mm were prepared on a 316 L stainless steel substrate with different process parameters, shown in Figure 2. After the experiment, the specimens manufactured by LPBF were removed from the substrate by electrical discharge machining.

**Table 2.** Process parameters of LPBF manufacturing K418.

| Samples NO. | Laser Power $P$, W | Scanning Speed $v$, mm/s | Hatch Spacing $H$, mm | Layer Thickness $t$, mm | Volumetric Energy Density $VED$, J/mm$^3$ | Relative Density η, % |
|---|---|---|---|---|---|---|
| 1 | 360 | 2400 | 0.08 | 0.05 | 37.5 | 94.78 |
| 2 | 360 | 2400 | 0.08 | 0.03 | 62.5 | 98.04 |
| 3 | 320 | 2800 | 0.07 | 0.02 | 81.6 | 99.94 |
| 4 | 360 | 2400 | 0.08 | 0.02 | 93.7 | 98.29 |

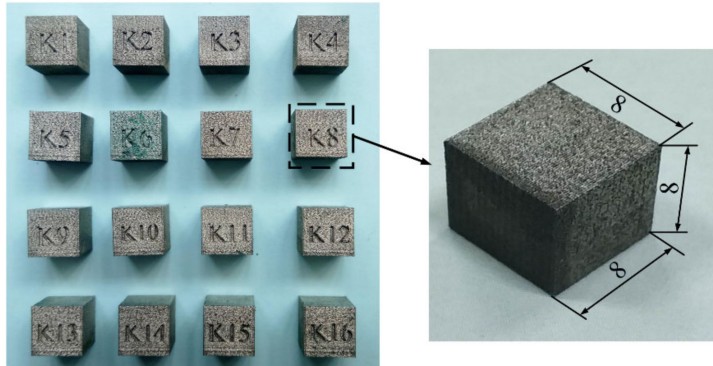

**Figure 2.** Macro photo of K418 sample manufactured by LFBF.

*2.3. Experimental Test*

The volumetric energy density *VED* was considered a comprehensive evaluation index to reflect the effects of the main process parameters of scanning speed *v*, laser power *P*, hatch space *H*, and layer thickness *t* on the microstructure and mechanical properties of LPBF-manufactured K418 samples. The volumetric energy density *VED* (J/mm$^3$) was defined as follows [21]:

$$VED = \frac{P}{v \cdot H \cdot t} \tag{1}$$

After LPBF, the surface of the as-built specimens was cut off by about 1 mm and mechanically polished with a diamond polishing agent for OM and SEM testing. To evaluate the effect of laser processing parameters on microstructure evolution, the prepared samples were ground with 400, 800, 1200, 1500, and 2000 mesh SiC metallographic sandpaper in sequence and then polished with a polishing fabric and a diamond polishing agent with a suspended particle size of 0.5 μm. Then, the cross-section of the polished samples was etched in aqua regia (HCl:HNO$_3$ = 3:1) for 60 s. optical microscopy (OM, Nikon, MA-200, Tokyo, Japan) was used to examine the micromorphology of the melting track and molten pool at low magnification. The microstructure between adjacent molten pools under magnification was observed by scanning electron microscopy (SEM, tescan VEGA, Brno, Czech Republic) equipped with energy-dispersive spectroscopy (EDS). The phase identification of K418 powder and LPBF-manufactured samples was performed by X-ray diffraction (XRD, model Bruker D8 Discover, Bremen, Germany) with Cu-Ka radiation (1.54 Å) at 40 kV and 40 mA. The crystal texture was observed by SEM-based electron backscatter diffraction (EBSD, Oxford Instruments, Abingdon, UK).

The as-built K418 tensile specimens were tested using a universal material testing machine INSTRON M4206at room temperature and a tensile rate of 1 mm/min in accordance with ASTM D638 GB/T 228-2010.

## 3. Results and Discussion

*3.1. Phases*

Figure 3 shows the XRD patterns of the LPBF-manufactured K418 samples with different volumetric energy densities *VED*. The γ-Ni phase is clearly identified as the only phase with the fcc crystal structure in the as-received powders. This may be attributed to the powder being obtained in an inert gas stream under nonequilibrium and extremely rapid cooling conditions, where the phase transformation process of the material is inhibited, and elements such as Cr, Fe, Mo are solid-dissolved into the Ni, resulting in a supersaturated solid solution of Ni γ (Ni-Cr-Fe) phase [22]. After the LPBF process, it was found that the LPBF-manufactured K418 samples mainly contained the higher diffraction peaks, which corresponded to the γ-Ni phase and γ'-Ni phase with the fcc crystal structure, and no carbides or other phases were detected. As can be seen in Figure 3. the diffraction peaks of the γ-phase and the γ' precipitate phase are substantially coincident with the diffraction angle ranges of 2θ = 20–120°. To distinguish the two existing phases accurately, XRD

diffraction analysis was observed by local magnification in a small range of 2θ = 50–52° and 2θ = 74–76°. As shown in Figure 3b, the 2θ position and the detected intensity of γ and γ′ show that the diffraction peaks of γ and γ′ expand significantly and that the intensity weakens significantly with decreasing volumetric energy density *VED*, which is primarily attributed to the imperfect crystal forms at a lower energy density. This can also reasonably explain and confirm the formation of clustered and fragmented microstructures in the case of a relatively low volumetric energy density *VED* of 37.5 J/mm$^3$ (as shown in Figure 3a). This may be attributed to the rapid cooling rate of 10$^5$–10$^8$ K/s during LPBF, resulting in a very short duration time (50–200 μm) for melting and solidifying the K418 powders, with most solute atoms, such as Nb, Mo, Cr, and Ti, being captured in the γ-Ni matrix, and it is difficult to undergo phase change [23]. Moreover, phase transformation is difficult to achieve in such a short period of time, and other phases, such as carbides, are also difficult to precipitate [24]. Actually, it is difficult to distinguish the γ′ phases rely solely on XRD from the matrix phase γ since their peaks overlap in Figure 3.

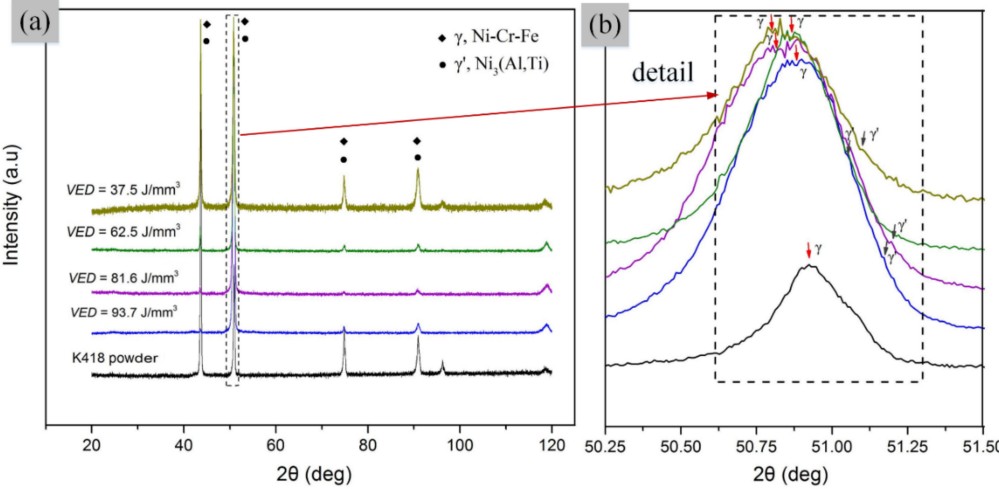

**Figure 3.** XRD of the LPBF-manufactured K418 with different *VED*: (**a**) 2θ = 20–120°; (**b**) 2θ = 50–52°.

The diffraction peaks 2θ tend to shift to the left when the volumetric energy densities *VED* decrease, as shown in Figure 3b and Table 3. The widening of the diffraction peaks may be attributed to the grain size, whereas the shifting in diffraction peak positions can be described by Bragg's law of diffraction [25]:

$$2d_{hkl} \sin \theta = n\lambda \tag{2}$$

$$d_{hkl} = a / \sqrt{h^2 + k^2 + l^2} \tag{3}$$

**Table 3.** Variation of diffraction peaks of γ-phase and γ′-phase under different energy densities.

| *VED* | γ-Phase | | γ′-Phase | |
|---|---|---|---|---|
| | 2θ/° | Intensity/CPS | 2θ/° | Intensity/CPS |
| 37.5 J/mm$^3$ | 50.78 | 1825 | 50.82 | 1900 |
| 62.5 J/mm$^3$ | 50.80 | 1788 | 50.85 | 1824 |
| 81.6 J/mm$^3$ | 50.83 | 1903 | 50.87 | 1926 |
| 93.7 J/mm$^3$ | 50.85 | 2185 | 50.90 | 2171 |

The grain size $d_{hkl}$ is proportional to the lattice constant, therefore, a larger $d_{hkl}$ results in a smaller 2θ. Therefore, the formation of supersaturated solid solutions of γ (Ni-Cr-Fe) and γ′ Ni$_3$ (Al, Ti) is responsible for the leftward shift of the diffraction peaks.

### 3.2. Microstructure

Figure 4 shows the OM topography of the K418 samples produced by LPBF using various parameters, revealing the influence of the volumetric energy density *VED* on the microstructure of the as-built K418 samples. It can be observed that different volumetric energy densities *VED* have a significant effect on the microstructure and morphology of K418 samples processed by LPBF. When the volumetric energy density *VED* is less than 62.5 J/mm$^3$, irregular columnar dendritic microstructures were obtained with significant considerable cracks and microvoids are observed. Especially when *VED* = 37.5 J/mm$^3$, there is an insufficient melting zone, and the microstructure of different regions differs significantly (Figure 4a). However, samples with an energy density higher than 81.6 J/mm$^3$ showed practically little crack failure (Figure 4c,d), the adjacent layers have good metallurgical bonding, the columnar crystal structure continuously runs through the multi-layers, and shows a preferential growth trend along the deposition direction. The microstructure at the interface between adjacent layers shows no discernible difference. At the same time, with the increase of the volumetric energy densities *VED*, so does the epitaxial growth tendency of columnar crystals.

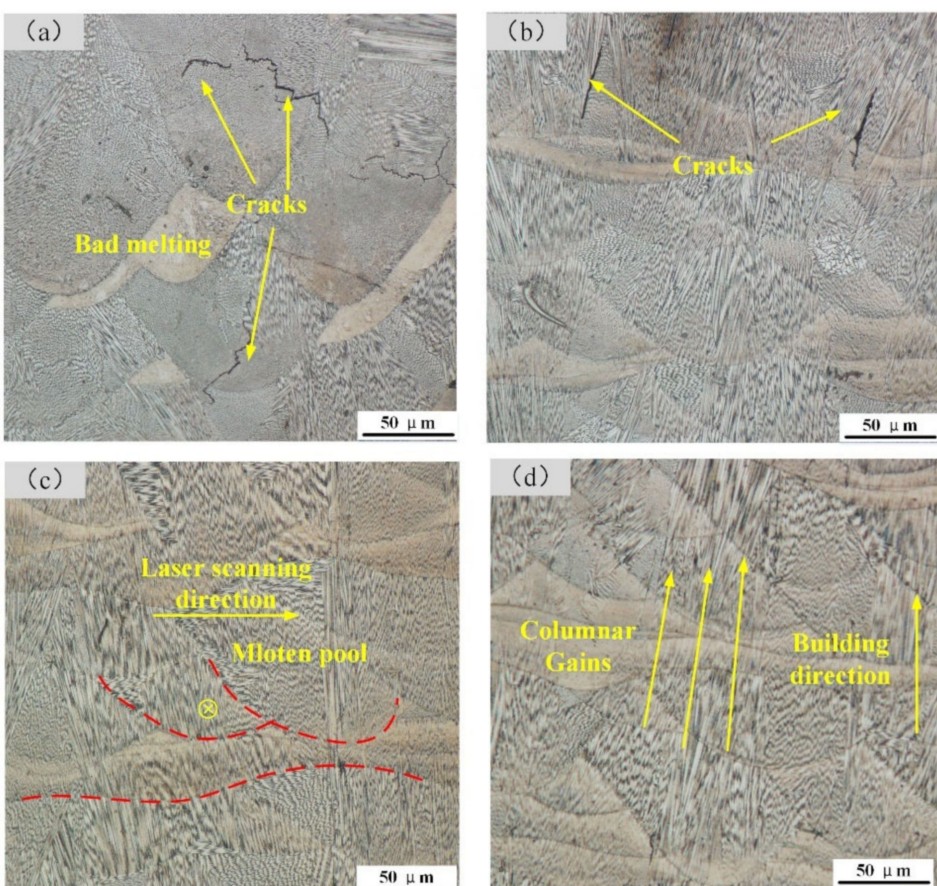

**Figure 4.** Optical micrographs of LPBF-manufactured K418 samples at various volumetric energy densities VED: (**a**) 37.5 J/mm$^3$; (**b**) 62.5 J/mm$^3$; (**c**) 81.6 J/mm$^3$; (**d**) 93.7 J/mm$^3$.

The SEM micrographs of the LPBF-manufactured K418 samples are shown in Figure 5. Continuous ultrafine columnar grains extending across multiple layers can be clearly observed in all samples with varying energy densities *VED*. There is no significant change in the interface between two neighboring layers, and the columnar grains are very continuous and uniform. However, as the *VED* increases from 62.5 J/mm$^3$ to 93.7 J/mm$^3$, the epitaxial growth of the continuous columnar grain becomes more pronounced. In the process of LPBF, the crystal growth perpendicular to the solid-liquid interface from the unmelted

substrate promotes crystal growth in the molten pool. The growth orientation of dendrites starts from the substrate and grows upward along the deposition direction. The heat flow direction of the substrate is parallel to the deposition direction. During the LPBF processing, the previously solidified layer is remelted, resulting in the continuous growth of dendrites through the multi-layer melting tracks. The rapid cooling of the molten pool leads to a strong non-equilibrium solidification structure and solute redistribution. The ultra-high temperature gradient within the molten pool leads to the rapid directional growth of crystals and the formation of columnar crystal structure. The mechanical properties of the LPBF-manufactured K418 sample are determined by the solidification microstructure, which basically depends on the local solidification conditions of the molten pool (solidification rate $V$, thermal gradient at the solid/liquid interface $G$) [26]. The material has experienced multiple complex thermal cycles. due to the melting layer and solidified previous layer during the LPBF process. The variation in the heat accumulation and the temperature evolution in the molten pool with different volumetric energy densities $VED$ results in a considerable difference in the internal energy and thermodynamic potential of the growth of columnar grains, leading to different driving forces for columnar dendritic nucleation and growth. Therefore, the $G/V$ value is the key parameter determining solidification structure characteristics.

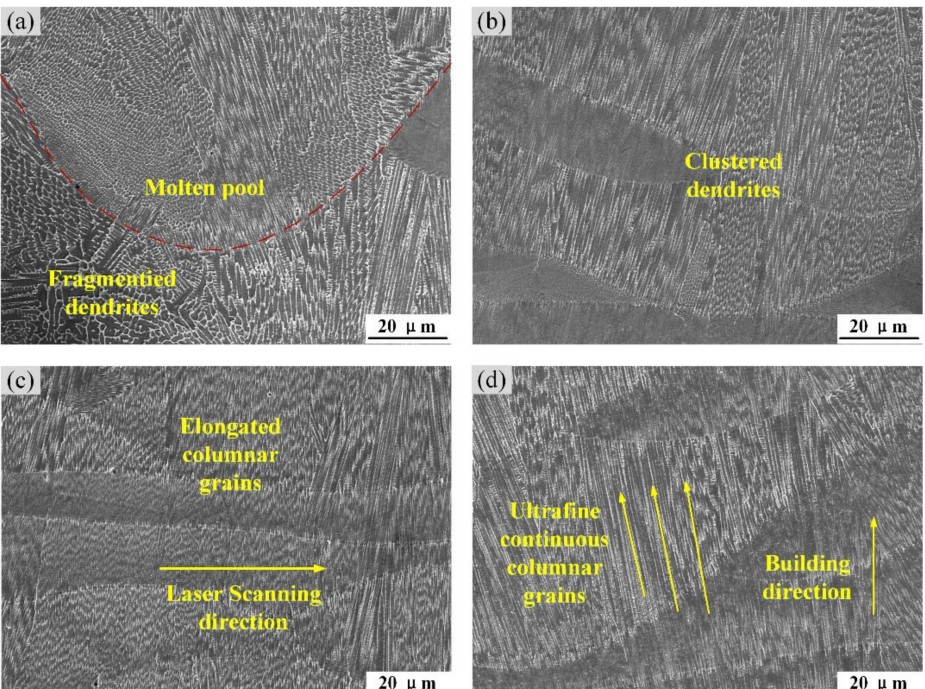

**Figure 5.** SEM micrographs of LPBF-manufactured K418 samples at various energy densities $VED$: (**a**) 37.5 J/mm$^3$; (**b**) 62.5 J/mm$^3$; (**c**) 81.6 J/mm$^3$; (**d**) 93.7 J/mm$^3$.

As shown in Figure 6, SEM and EDS scanning element distribution of the LPBF-manufactured K418 sample. It can be seen from Figure 6 that the elements of the LPBF-manufactured K418 samples are basically evenly distributed within and between crystals, and there is no obvious segregation. This is caused by rapid solidification and strong Marangoni convection in the molten pool. During LPBF processing, a large temperature gradient and transient solidification lead to significant grain refinement and a non-equilibrium solute capture effect, i.e., elements partitioning between gamma-matrix and precipitates, such as carbides, gamma prime, which effectively inhibits element segregation. At the same time, the strong Marangoni convection in the molten pool can further promote the uniform distribution of elements. It is widely assumed that the uniform distribution of elements in materials promotes uniform mechanical properties [27].

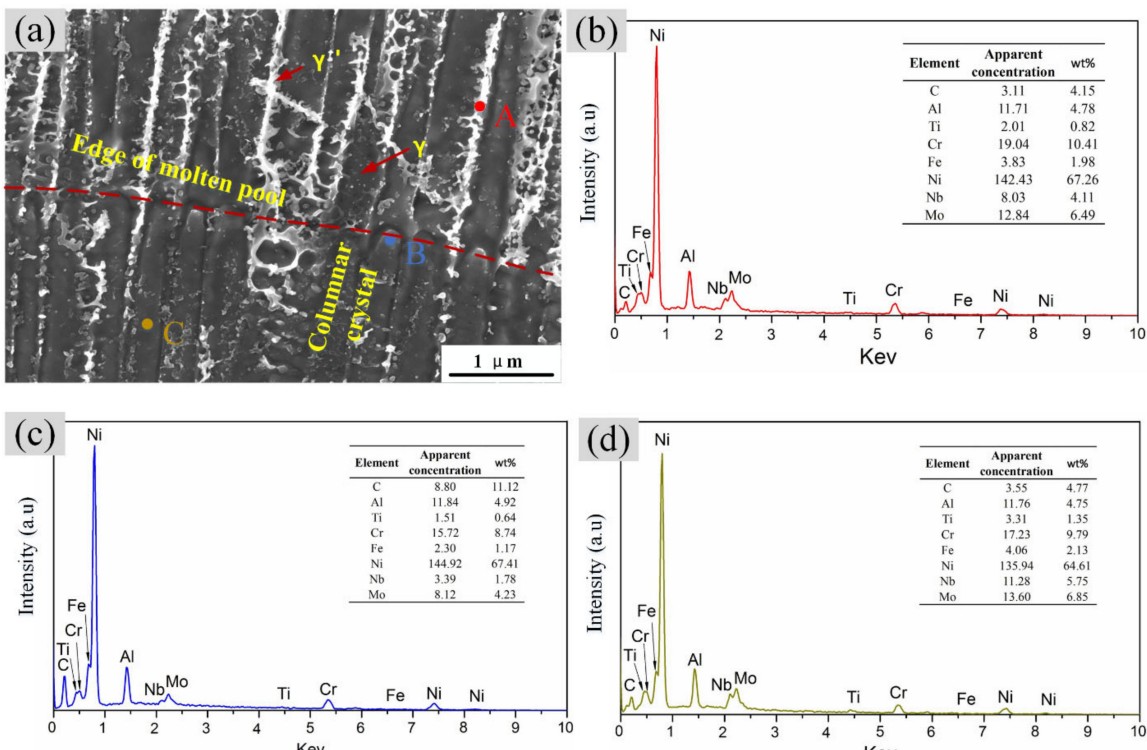

**Figure 6.** SEM and EDS scanning element distribution of the LPBF-manufactured K418 sample. (**a**) Columnar crystal morphology (**b**) Element composition of point A (**c**) Element distribution of point A (**d**) Element distribution of point A.

### 3.3. Texture

As shown in Figure 7, EBSD is performed on the LPBF-manufactured K418 samples at various energy densities. when the volumetric energy density *VED* is 37.5 J/mm$^3$, 81.6 J/mm$^3$, and 93.7 J/mm$^3$, the grains mainly tend to align along the (001) plane (red), are evenly distributed, and are arranged according to the (111) plane (blue) and (101) plane (green). At a volumetric energy density of 62.5 J/mm$^3$, the random distribution indicates that γ grains of K418 behave as a strong cubic texture {001} <100>. Nickel has a face-centered-cubic (fcc) crystal structure and columnar crystals that are typically aligned along <100> axes. During LPBF processing, the temperature gradient points from the center to the edge of the molten pool, and the rapid preferential growth of the Ni grains in the thermal gradient direction resulted in elongated columnar grains along the temperature gradient, i.e., the maximum heat flux direction [28].

When the energy density is relatively low, fragmented and incomplete columnar crystals are distributed around the molten pool, resulting in a weak texture and a slightly random orientation distribution. In the process of LPBF forming K418, the solidification generally starts from the edge of the molten pool and gradually grows towards the center of the molten pool. And along the deposition direction, the heat flux density is the largest, therefore, the solidified structure shows a <100> texture, which means that the preferential growth along the deposition direction and the tendency to grow preferentially is reinforced, therefore the crystallinity texture of the grains also strengthened. However, with a further increase in volumetric energy density *VED*, the temperature of the molten pool increases rapidly, resulting in intensive oxidation. Figure 8 is the ODF diagram of the three-dimensional orientation value of the φ = 45° cross-section of the K418 sample processed by LPBF [29]. As can be seen, the as-deposited K418 sample exhibits a cubic texture {110} <001>, which is determined by the preferred crystal orientation and maximum heat flow direction, which is also consistent with the crystallographic close-packed

plane <100> and <111> directions of the face-centered cubic nickel-based superalloy K418. Consequently, the solidified crystal texture tends to be in the <100> direction.

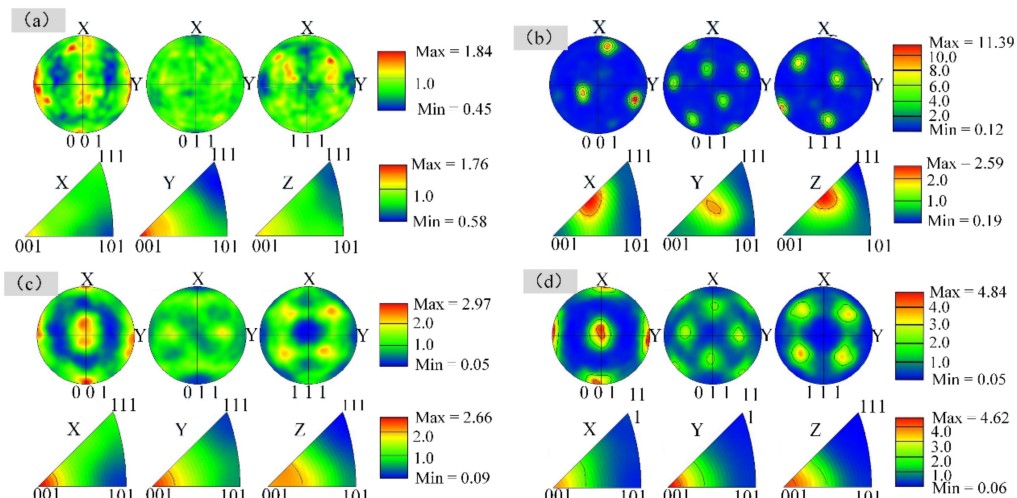

**Figure 7.** Crystal preferred orientations obtained through EBSD at various energy densities VED: (**a**) 37.5 J/mm³; (**b**) 62.5 J/mm³; (**c**) 81.6 J/mm³; (**d**) 93.7 J/mm³.

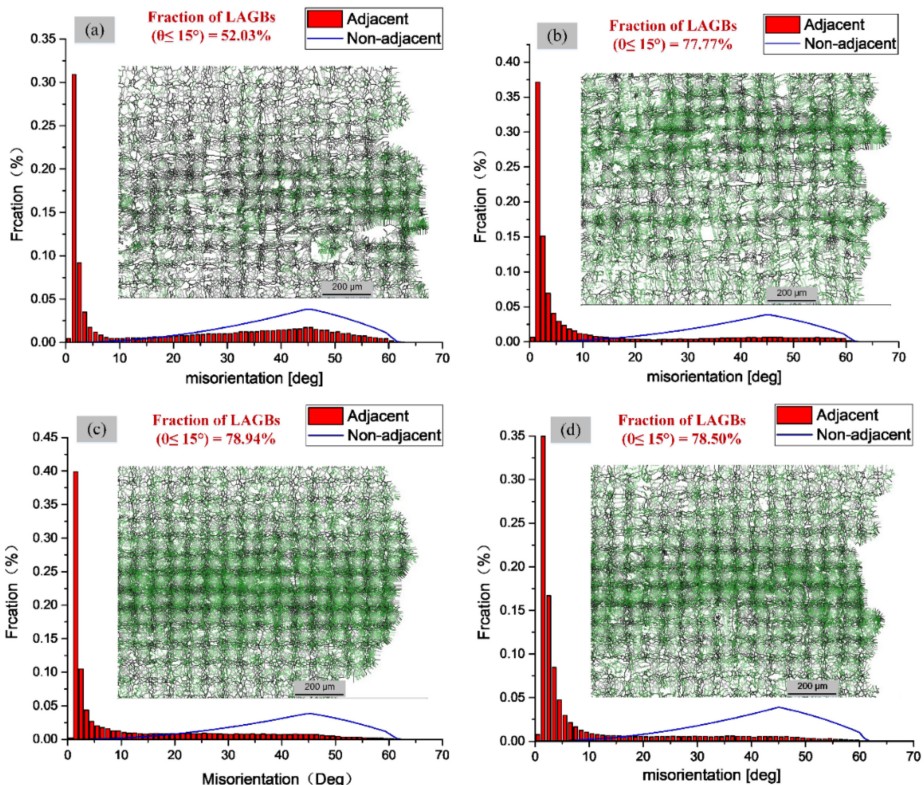

**Figure 8.** The misorientation distribution at various energy densities VED: (**a**) 37.5 J/mm³; (**b**) 62.5 J/mm³; (**c**) 81.6 J/mm³; (**d**) 93.7 J/mm³.

The results of the EBSD test are also reflected in the low-angle grain boundaries (*LAGBs*, θ ≤ 15°), Figure 8 depicts the grain boundary misorientation angle distribution at different energy densities, and the green lines indicate low-angle grain boundaries. As the volumetric energy density of *VED* increases from 37.5 J/mm³ to 81.6 J/mm³, the *LAGBs* were evenly distributed in all samples, and their fraction increased from 52.03% to 78.94%. However, further increasing the energy density *VED* to 93.7 J/mm³, a decrease rather than

an increase in the fraction of *LAGBs* (78.5%). *LAGBs* can bring high mechanical properties due to impeding the dislocation climb [30]. As illustrated in Figure 8, as the laser energy density *VED* increases from 37.5 J/mm$^3$ to 93.7 J/mm$^3$, the fraction of *LAGBs* increases initially and then decreases and reaches a maximum of 78.95% at a *VED* of 81.6 J/mm$^3$. This partly explains the different mechanical properties of K418 samples processed by LPBF under different energy densities. On the other hand, the grains of the LPBF-manufactured K418 samples are relatively fine, and the LAGBs content is generally higher, thus showing better mechanical properties than traditional casting.

### 3.4. Mechanical Properties

Figure 9 reveals the relationship between the microstructure and mechanical properties of LPBF-manufactured K418 with different power densities. When the volumetric energy density VED is less than 62.5 J/mm$^3$ with a high scanning speed, the duration of the interaction between the K418 powder and Gaussian laser source is relatively short, resulting in a lower bed temperature and a lower temperature gradient from the center to its side of the molten pool. In addition, the lower energy density induced by the higher scan speed suggests a relatively smaller $G/V$, thereby resulting in coarsened and imperfect columnar dendrites with cluster-like features. However, because of the lower temperature gradient at the lower volumetric energy density *VED* during LPBF, leading to lower surface tension, the fluid of the free surface was dragged to the edge of the molten pool at a relatively high speed. Meanwhile, the relatively low-temperature liquid within the molten pool flows to the center of the pool, creating anticlockwise and clockwise vortices in the molten pool. Marangoni convection is very weak in the molten pool, and the thermal conductivity is suppressed, preventing the K418 powder bed from melting completely and the solidified previous layer from melting, leading to defects such as pores and microcracks being generated at the layer interface.

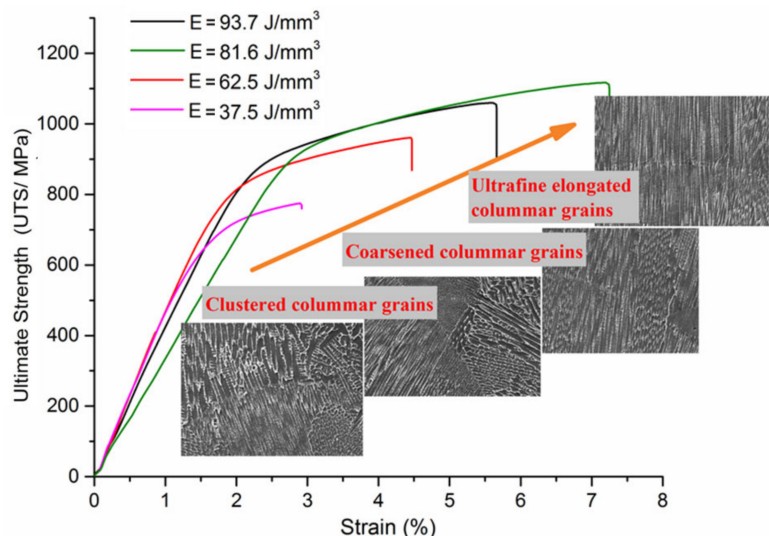

**Figure 9.** The relationship between microstructures and mechanical properties under different energy densities.

With decreasing laser power and scanning speed, the volumetric energy density increased to 81.6 J/mm$^3$, and the exposure time of the laser source on the powder particles and the molten pool was prolonged, leading to an increase in the powder particles' temperature, and the Marangoni convection promoted the wettability and expansion of the molten pool. As a result, reducing the heat transfer in the molten pool and extending cooling time provide sufficient dynamic solidification conditions for grain nucleation and growth. Because of the relative balance of the cooling rate at each position of the molten pool during extremely rapid cooling, due to sufficient nucleation and growth driving forces, the thermodynamic and potential internal energy produced by heat accumulation contribute to

the preferred epitaxial growth of columnar dendrites. As a result, the columnar dendrites exhibit ultrafine, continuous, and preferentially growth along the deposition direction, with no secondary dendrites being found. This is primarily due to the large $G/V$, resulting in the crystalline backbone being parallel to each other and growing along the maximum heat flux direction, which tends to coincide with the building direction, and lateral growth of the grains is suppressed completely.

The volumetric energy density further was increased to 93.7 J/mm$^3$, indicating that the K418 powder was irradiated under a higher energy intensity laser beam, causing the temperature of the molten pool in a local to rapidly rise above the melting point of nickel. As the laser source moves, the molten pool solidifies rapidly after the laser beam, and the crystal grains tend to grow toward the center of the molten pool along the moving direction of the laser source [31]. During the LPBF process, the heat flux and temperature gradient in the deposition direction are both the largest, therefore, the grain growth rate is also the fastest in this direction. The higher laser energy, causes a higher temperature gradient, ensuring that the underlying layer is completely melted, therefore, the epitaxial growth tendency of the columnar dendrites is gradually strengthened.

Figure 10 shows the Schmid factor distribution of LPBF-manufactured K418 samples at different energy densities. According to Schmid's law, the material yield strength can be calculated as follows [32]:

$$\tau_c = \sigma_s \cos \lambda \cos \varphi \tag{4}$$

where $\tau_c$ is the critical resolved shear stress in slip systems, which only depends on the nature of the material, $\cos \lambda \cos \varphi$ indicates the Schmid factor (SF), and $\sigma_s$ is the material yield strength. Thus, a lower Schmid factor leads to higher yield strength. All of the LPBF-manufactured K418 samples have a relatively small Schmidt factor, indicating that the as-built specimen has a high yield strength at various energy densities. Consequently, Figure 9 and Table 3 show that a higher yield strength was obtained at the optimized volumetric energy density *VED* of 81.6 J/mm$^3$, due to the relatively smaller SF of 0.4475. As a result, Schmid factor analysis further confirmed the effect of the volumetric energy density on the microstructure and mechanical properties of the LPBF-manufactured K418 samples.

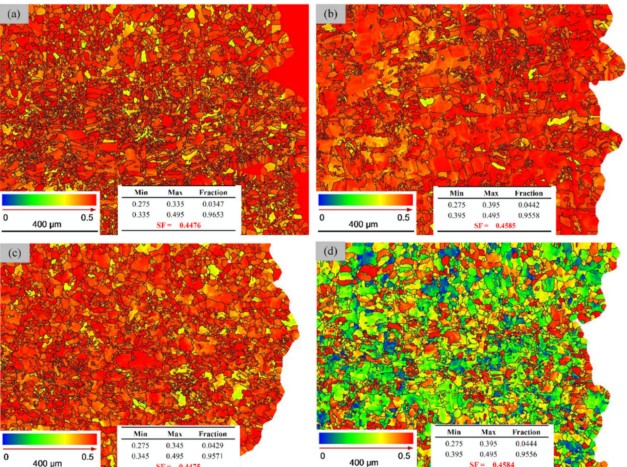

**Figure 10.** Schmid factor distribution at different volumetric energy densities VED: (**a**) 37.5 J/mm$^3$; (**b**) 62.5 J/mm$^3$; (**c**) 81.6 J/mm$^3$; (**d**) 93.7 J/mm$^3$.

As seen in Table 4, the ultimate strength and ductility of the LPBF-manufactured K418 samples under different energy densities are larger than those of the casting. The reason may be attributed to the rapid cooling rate and the high thermal gradient during the LPBF process.

**Table 4.** Mechanical properties of the LPBF-manufactured K418 samples at room temperature.

| | Ultimate Strength UTS/(Mpa) | Yield Strength YS(Mpa) | Elongation (%) | Grain Size (μm) | Aspect Ratio | Reference |
|---|---|---|---|---|---|---|
| As-cast | 875 | 735 | 7.6 | – | – | [1] |
| Sample 1 | 775 | 687 | 2.94 | 18.32 | 2.09 | This work |
| Sample 2 | 961 | 824 | 4.48 | 23.23 | 2.17 | This work |
| Sample 3 | 1117 | 912 | 5.67 | 21.53 | 1.94 | This work |
| Sample 4 | 1060 | 875 | 7.31 | 22.26 | 1.96 | This work |

Moreover, the yield strength $\sigma_s$ is generally inversely proportional to the grain size and can be calculated by the Hall–Petch relationship [33,34]

$$\sigma_s = \sigma_0 + \frac{K_S}{\sqrt{d_{mean}}} \tag{5}$$

where $\sigma_0$ and $K_S$ are constants, $\sigma_0$ indicates the lattice friction force generated when moving a single dislocation.

Grain boundaries are an obstacle to the slippage of dislocations according to dislocation theory. Owing to the stacking of dislocations, the grain boundaries can increase the stress when the stress becomes large enough to initiate the propagation of slip between adjacent grains [35]. Thus, the ultrafine columnar crystals result in a higher dislocation density, which impedes the movement of dislocations, thus strengthening the material. As exhibited in Figure 9 and Table 3, the samples with the volumetric energy density VED of 81.6 J/mm$^3$ display a smaller grain size (21.53 μm), corresponding to a higher ultimate tensile strength (1117 MPa) than the VEDs of 62.5 J/mm$^3$ (23.23 μm/961 MPa) and 93.7 J/mm$^3$ (22.28 μm/1060 MPa). However, the sample with a laser energy density VED of 37.5 J/mm$^3$ has a smaller grain size (18.32 μm) but a significant reduction in mechanical properties owing to the clustered and fragmented microstructures as well as considerable microcracks.

*3.5. Fracture Morphology*

As shown in Figure 11, the fractures morphology of the LPBF-manufactured samples under different energy densities. At a lower volumetric energy density VED lower of 37.5 J/mm$^3$, the lack of fusion, un-melted particles, and cracks on the fracture surface (Figure 11a) is a typical brittle fracture mode, hence resulting in a lower mechanical strength. While the samples with the volumetric energy density VED of 62.5 J/mm$^3$ (Figure 11b), some cleavage surfaces, cleavage steps, and ultrafine elongated columnar grains are observed, suggesting a ductile-brittle fracture mode. Some of the irregularly distributed micropores can be observed on the surface of the fracture, which acts as a crack initiator when the stress is concentrated, reducing the stress concentration zone, destroying the stability of the microstructure, and lowering the mechanical strength [36]. However, in samples with the VED above 81.6 J/mm$^3$ (Figure 11c,d), the fracture surface features a typical transgranular ductile fracture mode with fine shallow dimples, whose size appears to be similar to the width of the dendritic cells. It is encouraging that the fracture path does not follow adjacent molten pool boundaries, indicating good metallurgical bonding between adjacent layers [37]. Microcracks do not arise often and their propagation rate is low, consequently, excellent mechanical properties are obtained.

It can be seen from Figure 12 that the mechanical properties of the LPBF-formed K418 samples follow a similar trend with energy density to that of relative density, with a unimodal distribution that increases first and then decreases. In order to evaluate the comprehensive effects of microstructure characteristics (phase), crystal orientation (texture), low-angle grain boundaries (LAGBs), grain size, and defects (pores, cracks, etc.) on the mechanical properties of LPBF-manufactured K418 samples under different laser energy input, Figure 12 is divided into three regions: A, B, and C. In the A region, that is, $VED \leq 62.5$ J/mm$^3$, due to insufficient laser energy input, the relative density of the sample is low, there are significant internal defects such as microcracks and pores, and the

microstructure presents a fragmented and incomplete dendrite morphology, the crystal orientation (texture) is weak, and the main factors affecting the mechanical properties are defects and microstructure. In region B, i.e., when 62.5 J/mm$^3$ < E* < 81.6 J/mm$^3$, the laser energy input is sufficient to completely melt the powder and form a strong metallurgical bond between adjacent layers. The microstructure exhibits a preferred growth tendency across multi-layer melting tracks, showing a strong cubic texture <100>, there are more low-angle grain boundaries and small grain size, which comprehensively affects the mechanical properties of the sample. In the C region, that is, when $VED \geq 81.6$ J/mm$^3$, the powder absorbs excessive energy, resulting in defects such as over-burning and pores. The tendency of columnar crystals to grow directionally along the deposition direction is more obvious, while the grain size becomes larger and the amount of small-angle grain boundaries decreases, the main factors affecting the mechanical properties are microstructure, grain size, and structural defects.

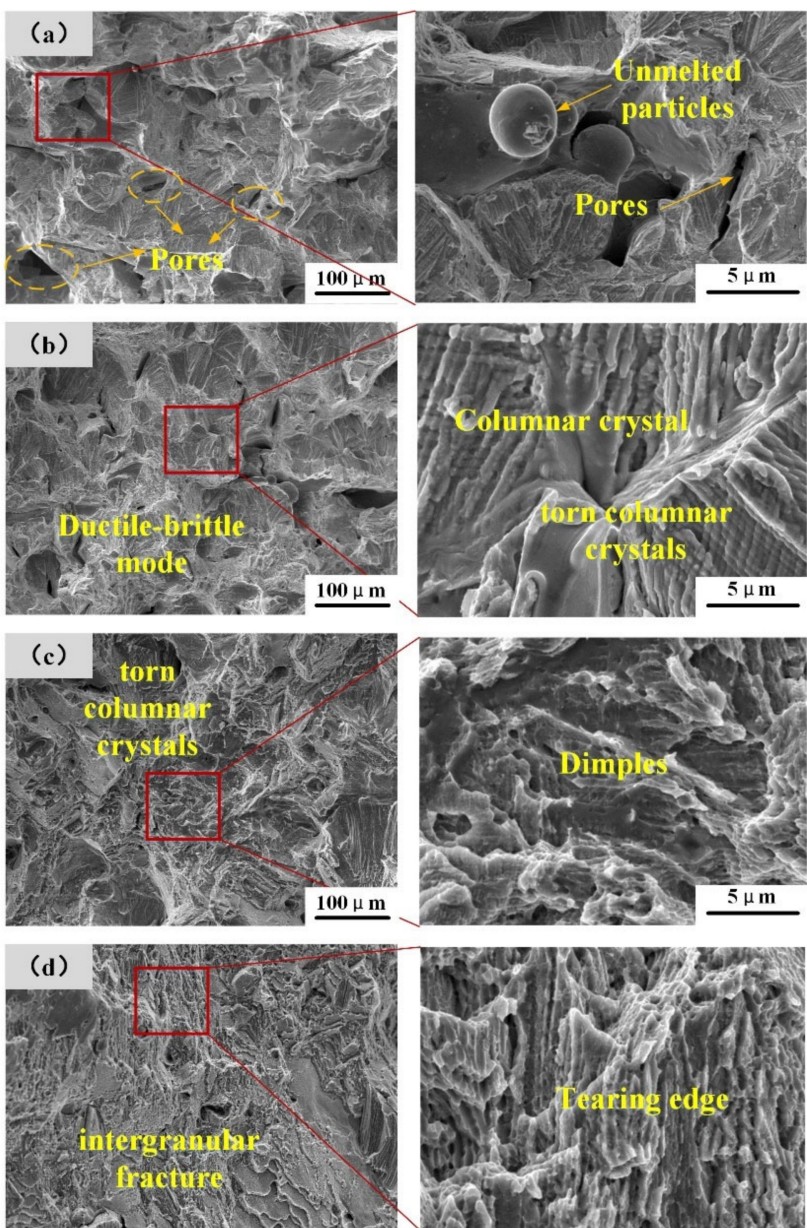

**Figure 11.** SEM images showing fracture surfaces for K 418 samples processed with LPBF at various energy densities VED: (**a**) 37.5 J/mm$^3$; (**b**) 62.5 J/mm$^3$; (**c**) 81.6 J/mm$^3$; (**d**) 93.7 J/mm$^3$.

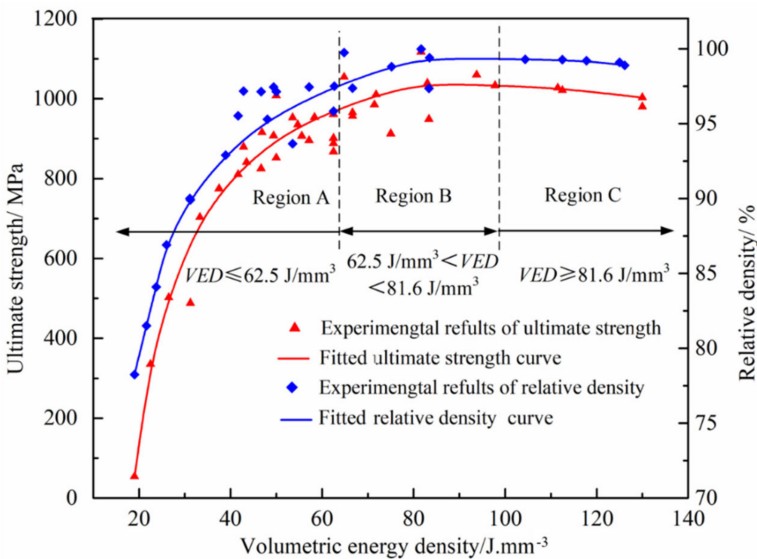

**Figure 12.** Effect of energy densities on mechanical properties.

## 4. Conclusions

In this work, the effect of process parameters on the microstructure, texture features, and mechanical properties of K418 samples processed with LPBF was investigated experimentally. The main findings can be summarized as follows:

1. (The mechanical properties of the LPBF-manufactured K418 sample are determined by the solidification microstructures, which are governed by the local solidification conditions ($G/V$) of the molten pool, with the value of $G/V$ being primarily controlled by the process parameters. As a result, the process parameters during LPBF are responsible for the evolution of the solidification microstructures and properties.

2. As the volumetric energy density increased, the typical microstructures of the LPBF-manufactured K418 samples evolved sequentially: clustered columnar grains→coarsened columnar grains→ultrafine elongated columnar grains.

3. As the volumetric energy density $VED$ increases from 37.5 J/mm$^3$ to 93.7 J/mm$^3$, the mechanical strength initially increases and then decreases reaching a maximum of 1117 MPa at the $VED$ of 81.6 J/mm$^3$, owing to the ultrafine elongated columnar grains and a strong {001} <100> cubic texture.

4. The fracture mechanism of the LPBF-manufactured K418 components changes from brittle fracture to ductile-brittle hybrid fracture to transgranular ductile fracture as the volumetric energy density $VED$ increases.

5. The solidification microstructure characteristics, grain orientation, low-angle grain boundaries, grain size, and internal defects comprehensively determine the final mechanical properties of the LPBF-manufactured K418 parts, and their respective influences on the properties are different under various process conditions.

**Author Contributions:** Methodology, Z.C.; Testing and characterization, Z.C., P.W., H.C., X.C., Y.R., W.Z. and S.L.; validation, Z.C. and P.W.; resources, Z.C. and S.L.; writing, Z.C. and P.W.; funding acquisition, Z.C. and S.L.; All authors have read and agreed to the published version of the manuscript.

**Funding:** This research is financially supported by the First China Special Assistant Postdoctoral Science Foundation (Grant No. 2019TQ0259) and the Fund of State Key Laboratory of Nickel and Cobalt Resources Comprehensive Utilization of Jinchuan Group (Grant No. GZSYS-KY-2020-016).

**Institutional Review Board Statement:** Not applicable.

**Informed Consent Statement:** Not applicable.

**Data Availability Statement:** Not applicable.

**Acknowledgments:** We gratefully thank the reviewers and editor. The authors at XJTU are grateful to the State Key Laboratory of Nickel and Cobalt Resources Comprehensive Utilization for funding this work.

**Conflicts of Interest:** The authors declare no conflict of interest.

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
