# Peer review of "Laser Powder Bed Fusion of K418 Superalloy: Process, Microstructure, Texture Feature, and Mechanical Property"

_metals, doi:10.3390/met12040611_

Round 1

Reviewer 1 Report

The article - is a good attempt to use well-known material for nowel manufacturing method. But, the necessity of this is not argued. You should give more proves that this work is necessary to do. Please, from this concern, improve the introduction. Pay high attention to the style of English. And use the same terms throughout the manuscript. Also, look through the pdf attached and make corresponding corrections. And please, add the final decision: may Your method be used to manufacture engine parts, or does it require additional studies? What may be improved in further research?

Author Response

Dear professor,

We sincerely appreciate all of your insightful comments and criticism. Those comments are all valuable and very helpful in modifying and improving our paper, as well as having important guiding significance for our research. We thoroughly reviewed your comments and revised them one by one; the revised sections are highlighted in the revised manuscript. The main corrections and responses are as follows:

(I) The article is a good attempt to use well-known material for novel manufacturing method. But, the necessity of this is not argued. You should give more proves that this work is necessary to do. Please, from this concern, improve the introduction.

Response:

Thanks for your professional and insightful guidance, we have supplemented the necessity of this study in the Introduction section.

K418 superalloy is a critical material in the production of hot-end components for aerospace engines. It is also an indispensable material for the manufacture of high-temperature hot end components for ships, automobiles, and gas engines. It is also widely used in a variety of industries, including transportation, the petrochemical industry, energy power, medical devices, and environmental protection. It can be used to manufacture aero-engine and gas turbine turbines, working blades, guide vanes, combustion chambers, and other hot end high-temperature components operating at 600-950°C due to its good high-temperature mechanical properties, economic outcomes and long-term structural stability, good casting performance, excellent thermal strength, thermal stability, and good resistance to mechanical fatigue and thermal fatigue. However, few studies have focused on K418 components fabricated by LPBF, which is largely due to the fact that K418 alloy, as a high-concentration γ'-strengthened superalloy, has a high content of Al and Ti elements, making the materially poor in weldability. The process window for LPBF manufacturing of K418 alloy is actually quite narrow and unreasonable process parameters can easily cause thermal cracks and rapidly deteriorate the performance of the material, making the engineering application of additively manufactured parts potentially risky and reliability difficult to guarantee. In addition, the cracking sensitivity and strength-plastic mismatch of LPBF manufactured K418 superalloy are the main technical bottlenecks preventing its use in high-strength and tough hot-end components.

(2)Pay high attention to the style of English. And use the same terms throughout the manuscript. Also, look through the pdf attached and make corresponding corrections.

Response:

We appreciate your enthusiastic guidance and time spent on our manuscript. We have made revisions one by one in response to your comments and suggestions, particularly the section you mentioned in the pdf, and the English of the full text has been polished, which are highlighted in the revised manuscript. Please see the attachment for details.

(3)And please, add the final decision: may Your method be used to manufacture engine parts, or does it require additional studies? What may be improved in further research?

Laser Powder Bed Fusion (LPBF) is regarded as one of the most potential technologies in metal additive manufacturing, it fabricates nearly full dense parts in a layer-by-layer manner by selectively melting and consolidation of thin layers of loose powder using the as-defined high-energy laser beam as an energy source. Due to its point-by-point complete melting mechanism, the LPBF exhibits a great potential for creating complex parts (such as overhang, curved surface, porous structure) that are used directly as end-use products, such as engine parts, gas turbine turbines, working blades, guide vanes, combustion chambers, and other hot end high-temperature components. The LPBF-manufactured parts with high precision have mechanical properties that are equivalent or superior to the wrought counterparts. K418 parts can be fabricated by LPBF with required microstructure and properties through process optimization. However, the as- built parts required support removal, surface polishing, and necessary heat treatment before they can be put into service.

We tried our best to improve the manuscript and made some changes to the revised manuscript. We try to avoid grammatical errors, and the English has been polished. These modifications should have no impact on the content and framework of the paper.

If you have any questions, please do not hesitate to contact us.

Reviewer 2 Report

Dear Authors,
I appreciate all your hard work. The article submitted by you is of high standard. The subject matter of the paper is timely.
The abstract is correct
The keywords are chosen correctly
Introduction - The authors write what research they do but there is no confirmation of why they do it. Please add few sentences which justify research done by Authors.
Very good description of the material prepared for specimens is worth mentioning.
In the method of research, please include a macro photo of the sample. Please state how many exposed layers the sample consists of. Please mark the area of the sample in the picture.
The LPBF optical microfractography is correct.
EBSD studies add to the significance of the work.

In microfractography studies, please provide a photograph of the sample. Please indicate the location where the SEM study was performed. Please indicate and describe the edge effect - if any. Please indicate the source of the fracture.

Kind regards

Author Response

Dear professor,

We sincerely appreciate all of your insightful comments and criticism. Those comments are all valuable and very helpful in modifying and improving our paper, as well as having important guiding significance for our research. We thoroughly reviewed your comments and revised them one by one; the revised sections are highlighted in the revised manuscript. The main corrections and responses are as follows:

(1)Introduction - The authors write what research they do but there is no confirmation of why they do it. Please add few sentences which justify research done by Authors.

Response:

Thanks for your professional and insightful guidance, we have supplemented the necessity of this study in the Introduction section.

At present, considerable work regarding the microstructures and properties of nickel-based superalloys has been carried out, such as Inconel 625 [15], Inconel 718 [16, 17], Hastelloy-X [18], Waspaloy [19], and IN738LC [20]. K418 superalloy is a critical material in the production of hot-end components for aerospace engines. It is also an indispensable material for the manufacture of high-temperature hot end components for ships, automobiles, and gas engines. It is also widely used in a variety of industries, including transportation, the petrochemical industry, energy power, medical devices, and environmental protection. It can be used to manufacture aero-engine and gas turbine turbines, working blades, guide vanes, combustion chambers, and other hot end high-temperature components operating at 600-950 °C due to its good high-temperature mechanical properties, economic outcomes and long-term structural stability, good casting performance, excellent thermal strength, thermal stability, and good resistance to mechanical fatigue and thermal fatigue. To date, the influence of casting, forging, and injection molding on the microstructure and mechanical properties of the K418 alloy has been studied extensively. However, few studies have focused on K418 components fabricated by LPBF, which is largely due to the fact that K418 alloy, as a high-concentration γ'-strengthened superalloy, has a high content of Al and Ti elements, making the materially poor in weldability. The process window for LPBF manufacturing of K418 alloy is actually quite narrow and unreasonable process parameters can easily cause thermal cracks and rapidly deteriorate the performance of the material, making the engineering application of additively manufactured parts potentially risky and reliability difficult to guarantee. In addition, the cracking sensitivity and strength-plastic mismatch of LPBF manufactured K418 superalloy are the main technical bottlenecks preventing its use in high-strength and tough hot-end components.

(2)In the method of research, please include a macro photo of the sample. Please state how many exposed layers the sample consists of. Please mark the area of the sample in the picture.

Response:

We appreciate your comments and suggestions, we have added the macro photos of the sample and related information.

Cubical specimens with a size of 8 mm × 8 mm × 8 mm were prepared on a 316 L stainless steel substrate with different process parameters, the as-built specimens were removed from the substrate by electrical discharge machining. After LPBF, the surface of the specimens was cut off by about 1mm and mechanically polished with a diamond polishing agent for OM and SEM testing.

(3)In microfractography studies, please provide a photograph of the sample. Please indicate the location where the SEM study was performed. Please indicate and describe the edge effect - if any. Please indicate the source of the fracture.

Response:

We appreciate your comments and suggestions. In this manuscript, the dog-bone tensile samples with a gauge length of 25 mm were prepared according to ASTM D638 GB/T 228-2010, and then the mechanical performance testing of the as-built K418 tensile specimens was carried out using a universal material testing machine INSTRON M4206at room temperature and a tensile rate of 1 mm/min. After the experiment,

SEM was used to examine the fracture morphology of the tensile specimen. The fracture’s cross-sectional area was small, and the test was performed near the center of the fracture, the precise test position was not recorded during the experiment. However, the experimental results can well reflect the fracture morphology and fracture mechanism under different energy densities, and will not affect the correctness of the experimental results.

We tried our best to improve the manuscript and made some changes to the revised manuscript. We try to avoid grammatical errors, and the English has been polished. These modifications should have no impact on the content and framework of the paper.

If you have any questions, please do not hesitate to contact us.
